# Celiac Disease, Gluten-Free Diet and Metabolic Dysfunction-Associated Steatotic Liver Disease

**DOI:** 10.3390/nu16132008

**Published:** 2024-06-25

**Authors:** Georgiana-Diana Cazac, Bogdan-Mircea Mihai, Gabriela Ștefănescu, Georgiana-Emmanuela Gîlcă-Blanariu, Cătălina Mihai, Elena-Daniela Grigorescu, Alina Onofriescu, Cristina-Mihaela Lăcătușu

**Affiliations:** 1Unit of Diabetes, Nutrition, and Metabolic Diseases, Faculty of Medicine, “Grigore T. Popa” University of Medicine and Pharmacy, 700115 Iași, Romania; georgiana-diana.cazac@d.umfiasi.ro (G.-D.C.); alina.onofriescu@umfiasi.ro (A.O.); cristina.lacatusu@umfiasi.ro (C.-M.L.); 2Clinical Center of Diabetes, Nutrition and Metabolic Diseases, “Sf. Spiridon” County Clinical Emergency Hospital, 700111 Iași, Romania; 3Unit of Medical Semiology and Gastroenterology, Faculty of Medicine, “Grigore T. Popa” University of Medicine and Pharmacy, 700115 Iasi, Romania; gabriela.stefanescu@gmail.com (G.Ș.); georgiana.gilca@gmail.com (G.-E.G.-B.); catalina.mihai@umfiasi.ro (C.M.); 4Institute of Gastroenterology and Hepatology, “Sf. Spiridon” County Clinical Emergency Hospital, 700111 Iași, Romania

**Keywords:** celiac disease, gluten, gluten-free diet, non-alcoholic fatty liver disease, liver steatosis, MASLD

## Abstract

Celiac disease (CD) is a chronic autoimmune disorder triggered by the ingestion of gluten-containing food by genetically predisposed individuals. Hence, treatment of CD consists of permanent avoidance of wheat, rye, barley, and other gluten-containing foods. Lifelong adherence to a gluten-free diet (GFD) improves the symptoms of CD, but recent evidence suggests it is also associated with a higher risk for hepatic steatosis and the coexistence or emergence of other cardiometabolic risk factors. Moreover, a higher risk for liver steatosis is also reported by some authors as a potential extraintestinal complication of the CD itself. Recent nomenclature changes designate the association between hepatic steatosis and at least one of five cardiometabolic risk factors as metabolic dysfunction-associated steatotic liver disease (MASLD). An extended network of potentially causative factors underlying the association between MAFLD and CD, before and after dietary therapy is implemented, was recently described. The individualized treatment of these patients is less supported by evidence, with most of the current recommendations relying on empiric clinical judgment. This review focuses on the causative associations between CD and hepatic injury, either as an extraintestinal manifestation of CD or a side effect of GFD, also referring to potential therapeutic strategies for these individuals.

## 1. Introduction

Celiac disease (CD) is an immune-mediated enteropathy triggered by the consumption of gluten-containing foods in genetically predisposed individuals [1,2]. CD is characterized by a persistent intolerance to gluten, an essential protein in wheat, and to other gluten-like proteins found in rye (secalin), barley (hordein), or oats (avenins) in some cases, and features systemic and gastrointestinal manifestations, among which most are induced by autoimmune mechanisms [1,3]. CD predominantly involves the proximal small intestine, leading to malabsorption. The primary treatment for CD consists of strict adherence to a permanent gluten-free diet (GFD) [4,5,6].

CD is the most common lifelong food-related disorder worldwide, affecting nearly 1% of the global population, but most of these individuals remain undiagnosed due to the high variability of clinical signs and symptoms at presentation [7,8]. Epidemiological data worldwide show a higher prevalence depending on the prevalence of HLA genes and lifestyle [7,9]. In the United States, the relative prevalence of CD is associated with indicators of economic development. Regional sociodemographic variables such as median income and urban area were negatively correlated with Latino ethnicity, black race, and median social deprivation index score [10]. Improvements in diagnostic equipment and health systems in low-income countries, the ability to diagnose children without duodenal biopsy, and the expansion of Westernized eating habits in Asia have all contributed to the increased population that is diagnosed with CD worldwide [11,12]. The variability of screening practices between different countries points towards a need to develop universal guidelines able to cover both the rapid growth of the CD population and the corresponding gaps in the medical care health systems provided for these patients [12].

Even though CD is primarily acknowledged as an enteropathy, it is also associated with extraintestinal manifestations, some of which were only identified in recent years [13]. The American Association for the Study of Liver Diseases (AASLD) Practice Guidance lists CD as a less common cause of hepatic steatosis and steatohepatitis, which needs to be excluded when additional signs and symptoms like iron or vitamin D deficiency, diarrhea, abdominal pain, and bloating are present [14]. Several studies described frequent weight modifications after starting a GFD [15], an increased risk of metabolic features [16], and the development of hepatic steatosis as hallmarks of metabolic syndrome (MS) [17]. However, the evidence is still contradictory. On one hand, some research suggests that gluten is a causal factor for hepatic injury during CD evolution [18,19]. On the other hand, other studies argue that following a lifelong GFD to treat CD can improve the symptoms but display a higher risk of determining some features of metabolic dysfunction-associated steatotic liver disease (MASLD) [20].

This article aimed to summarize the relationship between CD, adherence to GFDs, potential factors and mechanisms leading to MASLD features, as well as the distinctive features of the therapeutic approach of these patients with CD.

## 2. Celiac Disease and Metabolic Dysfunction-Associated Steatotic Liver Disease

### 2.1. Celiac Disease

As an autoimmune disease that evolves in genetically predisposed subjects, CD is commonly clustered in families and is characterized by the presence of specific HLA types, among which HLA-DQ2 and HLA-DQ8 are seen in almost all cases [2,21]. There are still many unanswered questions on how genetic and environmental factors are involved and interact in the development of CD [22]. Besides the role adaptive and innate immune responses play in CD pathogenesis, other probable causes include the early introduction of gluten, gastrointestinal infections, and alterations in the intestinal microbiota [23,24,25]. Environmental factors, such as timing, amount, or balance with breastfeeding, are primarily linked to the moment gluten is introduced during food diversification [2,22]. The CD is often associated with other autoimmune conditions like type 1 diabetes mellitus, thyroid diseases, Addison’s disease, inflammatory bowel disease, or autoimmune hepatitis (AIH) [26]; these maladies must be screened in celiac patients.

CD exhibits a wide range of gastrointestinal and extraintestinal symptoms, the latter including liver pathology such as steatosis and/or elevated transaminase levels [11]. The diagnosis of CD in adults requires a positive serology for immunoglobulin A (IgA), anti-transglutaminase 2 (TG2), and total IgA, along with evidence of villous modifications on a small intestine biopsy, such as varying degrees of villus atrophy, mucosal inflammation, and crypt hyperplasia [2,27]. In children, a biopsy can be avoided if their IgA anti-TG2 level exceeds 10 times the upper normal limit and they have positive IgA anti-endomysial (EMA) antibodies from an additional blood sample [27]. Treatment of CD can lead to clinical and histologic improvement in most patients [11,28]. Patients newly diagnosed with CD need close follow-up under treatment to monitor symptom improvement and dietary adherence. As about 20% of patients remain symptomatic due to noncompliance or poor understanding of the diet, it is recommended that new patients work with a dietitian to better understand dietary restrictions [25,29].

### 2.2. MASLD

Non-alcoholic fatty liver disease (NAFLD), as it was named until recently, has become the most common cause of liver disease worldwide, evolving alongside the global obesity epidemic and actually representing the hepatic manifestation of the metabolic syndrome [14,30]. According to data collected between 1990 and 2019 from all global regions, the prevalence of metabolic liver disease reaches approximately 30% of the world population. This increased prevalence is closely associated with the current pandemic rates of obesity and metabolic syndrome; therefore, the metabolic fatty liver is becoming the most frequent form of chronic liver disease and a public health challenge [31,32].

NAFLD is characterized by the hepatic accumulation of excessive fat in the absence of excessive alcohol consumption, virus C chronic hepatitis, or the use of drugs that can induce steatosis or other etiologies. This umbrella term was used to describe the whole histological spectrum, from steatosis to steatohepatitis and further to advanced fibrosis, cirrhosis, and hepatocellular carcinoma [33,34]. Not all individuals with NAFLD progress to more severe stages, but the coexistence of obesity, diabetes, and metabolic syndrome increases this risk. Therefore, early detection and management to prevent progression is essential, as early stages of NAFLD are often reversible with lifestyle changes [35,36].

NAFLD labeling has gone through several changes over time [35,37]. The new terminology and diagnostic criteria published in June 2023, aiming for non-stigmatization and improved patient awareness and identification, are widely supported. Based on these changes, non-alcoholic fatty liver disease (NAFLD) has become metabolic dysfunction-associated steatotic liver disease (MASLD), and non-alcoholic steatohepatitis (NASH) has become metabolic dysfunction-associated steatohepatitis (MASH) [14,38]. Studies suggest an almost complete overlap (99%) between the population defined as having MASLD and the population previously defined as having NAFLD. All recommendations on the clinical evaluation and management of NAFLD covered by the American Association for the Study of Liver Diseases (AASLD) Practice Guidelines can be applied to patients with MASLD and MASH. Results of natural history and biomarker validation studies in patients with NAFLD and NASH are considered valid for patients with MASLD and MASH, respectively, until further guidance [38,39,40]. Histologically or imaging-diagnosed steatotic liver disease (SLD) covers various potential etiologies. MASLD, defined as the presence of hepatic steatosis in combination with cardiometabolic risk factors and no other discernible cause, alcoholic liver disease (ALD), and an overlap of these two (MetALD), comprise the most common causes of SLD [38,39]. Given their distinct pathophysiology, other causes of SLD should be considered separately, as already accustomed in clinical practice. At the same time, the coexistence of other forms of liver disease with MASLD is possible, such as the association between MASLD and autoimmune hepatitis or viral hepatitis [38,39]. The diagnosis of metabolic liver disease is based on the presence of metabolic criteria and the exclusion of other causes of hepatic steatosis. The five cardiometabolic criteria are represented by either modified values of the body mass index (BMI) and fasting serum glucose, hypertension, high plasma triglycerides, and low HDL-cholesterol levels or an already existing treatment for these items [39]. In the presence of hepatic steatosis, detecting any cardiometabolic risk factor would lead to a diagnosis of MASLD if no other causes of hepatic steatosis exist. If additional pro-steatotic factors are identified, this is consistent with an overlapped etiology. In the case of alcohol-consuming patients, the liver disease is called MetALD or ALD, depending on the level of alcohol consumption. In the absence of manifest cardiometabolic criteria, other etiologies must be excluded, and if none are identified, the term cryptogenic SLD should be used. In advanced fibrosis or cirrhosis, steatosis may be absent, thus requiring a clinical judgment based on cardiometabolic risk factors and the absence of other etiologies [38,39,41].

### 2.3. The Link between Celiac Disease and Liver Injury

While for a long time CD was usually associated with undernutrition and low weight, many patients with CD (approximately 22% to 82%) are now found to be overweight or even have obesity, especially if on a GFD, which increases their risk of developing MS and SLD [42]. The evolution into MS is gradual, and its rate depends on the duration of the disease [42].

In the early 2000s, several reports focused on the incidence of elevated transaminases and liver disease in patients diagnosed with CD. Many of these studies found that subjects with cryptogenic hypertransaminasemia had asymptomatic CD identified by positive antigliadin and anti-endomysial antibodies and confirmed with duodenal endoscopy and biopsy, thus concluding that patients with chronically elevated liver enzymes should be screened for CD [43,44,45,46]. Moreover, it appeared that transaminase levels improved or even normalized after introducing a GFD [43,44]. In the case of liver enzyme elevation in patients with CD, hypertransaminasemia may be a sign of an associated liver disease or an epiphenomenon, especially at diagnosis and before starting a GFD, when intestinal permeability is increased [18]. As an extraintestinal autoimmune disease sometimes associated with CD, AIH can be a cause of hepatic injury that supports the potential immune-mediated pathogenesis of hypertransaminasemia in celiac patients [18,47]. Anti-actin antibodies in CD patients, which show a significant correlation with villous atrophy, also have a very high specificity in AIH [48,49]. Some authors believe it would be helpful to analyze whether these autoantibodies can represent liver damage markers [47].

In a study of 463 celiac patients, 40.6% had elevated ALT or AST levels at the time of CD diagnosis, compared to 24.2% of the already treated CD patients (*p* < 0.001) and 16.6% of matched controls (*p* < 0.001). Similarly, 36.7% of CD patients in the National Health and Nutrition Examination Survey (NHANES) database had abnormal ALT values, compared to 19.3% of non-celiac patients (*p* = 0.03). Approximately 79% of CD patients with elevated liver function tests at diagnosis normalized their laboratory data after a GFD was implemented for a mean duration of 1.5 ± 1.5 years [50]. While CD accounts for 6–9% of all high liver enzyme cases, hypertransaminasemia occurs in 40% of CD patients. Consequently, it is recommended that patients diagnosed with CD undergo hepatic enzyme screening [51].

When screening for CD in patients with NAFLD, Rahimi et al. found 2.2% positive cases, suggesting the need to screen for CD in individuals with NAFLD and no other associated condition [52]. Bakhshipour et al. also found a higher prevalence of CD in patients with NAFLD (3.22%) compared to the general population and issued a similar suggestion of screening patients with NAFLD for CD, referring once again to the causative role of CD in SLD [53].

Abdo et al. showed that CD can be associated with various forms of liver disease with abnormal liver enzyme levels, among which are primary biliary cirrhosis, primary sclerosing cholangitis, AIH, steatotic liver, or even end-stage liver disease. In these cases, the authors suggested that screening for CD and, if present, implementing a GFD are expected to improve the elevated liver enzyme levels [54]. Abenavoli et al. believe that steatotic liver can appear because of gluten-dependent hepatic involvement in patients with CD [55]. Gluten restrictions seem to reduce the risk of some CD manifestations and complications (e.g., malabsorption, osteoporosis, or even malignancy) in the group of patients who also have autoimmune liver diseases [56]. The European Society for Clinical Nutrition and Metabolism (ESPEN) Guidelines recommend initiating a GFD in patients with CD and NAFLD/NASH to improve liver enzymes and histology, prevent progression to cirrhosis, and improve intestinal pathology [56].

After Akpinar et al. found high levels of pigment epithelium-derived factor (PEDF) in subjects with CD and analyzed their effects on angiogenesis, Dikker et al. correlated these results with other studies that investigated the levels of PEDF in subjects with hepatic steatosis, arguing that both diseases occur on a common inflammatory background [57,58], and suggested that patients with CD and high levels of PEDF need further investigation for the presence of steatosis and vice versa [58].

## 3. Gluten-Free Diet and MASLD

MASLD is increasingly observed in people with CD who adhere to a GFD. Several pathophysiological mechanisms may contribute to the development of MASLD in this dietary context. These mechanisms intertwine metabolic, inflammatory, and immunological pathways, eventually leading to liver disease. Therefore, understanding the pathophysiology of MASLD has become an essential requirement, in addition to diagnosing and managing patients with CD at the right time. For now, the exact mechanism of MASLD’s association with CD is still unknown, although some hypotheses have been raised.

### 3.1. Nutritional Imbalance and Weight Gain

Following a GFD reduces the extraintestinal symptoms that often occur before diagnosis, such as anemia, weight loss, or fatigue. Anthropometric data such as BMI or underweight improve after gluten is removed from the diet of children or adults with CD [59,60,61].

Even though GFD is safe and shows a reduced risk of complications, some reports suggest that metabolic syndrome and hepatic steatosis may develop in patients with CD following a GFD [17,62]. On the contrary, other studies have shown that following a GFD can have some benefits in ameliorating hepatic steatosis [4,63,64,65]. These conflicting results require supplementary investigation to uniformize the view on this matter.

Adopting a GFD is not a lifestyle choice for patients with CD. As GFD is a restrictive diet, it can result in adverse nutritional effects, similarly to other dietary restrictions, especially over the long term. It is widely recognized that GFDs can be nutritionally insufficient and are often linked to vitamin and mineral deficiencies. Moreover, they are associated with high sugar and fat intake, mainly when gluten-free alternatives are used [66,67]. Some studies reported an increased risk of abnormal serum lipid levels, particularly triglycerides, in patients with CD on a GFD [42,68]. Regardless of serum lipid levels, numerous reports have noted an elevated risk of fatty liver and MASLD, potentially leading to a higher cardiovascular risk in these patients [68,69]. Intestinal absorption and some nutrient deficiencies seen in CD generally improve during a GFD, resulting in a favorable increase in HDL-cholesterol levels. As inflammation decreases, many metabolic parameters can normalize, although the levels of total cholesterol and triglycerides often rise [70]. Nevertheless, some nutrient deficiencies, such as proteins, fibers, group B vitamins, iron, folate, calcium, and magnesium, may be accentuated during a GFD, which can lead to metabolic dysfunction and increased oxidative stress in liver cells, thus promoting liver damage and inflammation [66,71].

Traditional gluten-free products (oats, glucose syrup, starch, and flour derived from rice) bring along an excess of fructose and carbohydrates that further determine a high-calorie intake and may increase the risk of MASLD [72]. Chronic intake of high-calorie foods leads to a positive energy balance, promoting adiposity and obesity. Besides the GFD composition by itself, this malabsorption disorder may often be followed by a hyperphagic compensatory response, leading to weight gain in patients with CD [20]. The excessive adipose tissue releases increased amounts of free fatty acids (FFAs) in the bloodstream, which are taken up by the liver and contribute to hepatic steatosis [55]. Obesity and adiposity enhance insulin resistance and create a pro-inflammatory state, contributing to the progression from simple steatosis to MASLD [73]. Many studies reported a rising frequency of body weight changes in patients with CD following a GFD [15,74,75]. As a result, individuals with CD tend to acquire higher rates of overweight, obesity, and metabolic complications, including steatotic liver and cardiovascular diseases [17,76]. The fact that GFD is low in fibers (due to reduced consumption of grains) seems even more detrimental to preventing these cardiometabolic conditions [77].

A prospective study on 26,861 individuals with CD and no modifications or history of liver disease at diagnosis found that patients with CD had a significant risk of developing NAFLD compared to the general population. The researchers underlined that the highest risk was in the first year after diagnosis, persisted over the 15 following years, and was significantly higher in the male population (male HR = 4.5 vs. female HR = 2.1) [78]. Following a GFD in patients with CD seems to determine features of MS by gaining weight, which leads to insulin resistance and a worsening glucose tolerance and lipid profile, with the development of hepatic steatosis as the hepatic hallmark of MS [62,79].

The assessment of the MS prevalence in patients with CD after starting a GFD showed that 29% of the 98 newly diagnosed patients enrolled in a study developed metabolic syndrome after one year of GFD. Therefore, compared to baseline, 72 vs. 48 subjects significantly increased their waist circumference, 18 vs. 4 had hypertension, 25 vs. 7 had higher glucose levels, 16 vs. 7 patients had hypertriglyceridemia, and 34 vs. 32 had lower levels of HDL-cholesterol. Later during follow-up, 28 patients had hepatic steatosis vs. 18 at baseline, although this was not a statistically significant difference [62].

In another study, researchers compared 202 patients with CD to 202 control subjects, concluding that over one-third of patients with CD adhering to a GFD also had concurrent NAFLD, hence resulting in a three-fold increased risk compared to the general population. Notably, the study concluded that the relative risk of NAFLD was significantly higher among non-overweight individuals with CD [20]. Rispo et al. also described an increased risk of MAFLD in CD patients following a GFD. Their retrospective study on newly diagnosed patients with CD assessed the prevalence of NAFLD and MAFLD in patients with CD at the time of diagnosis and after two years of GFD. The only significant difference that was induced by the use of NAFLD and MAFLD criteria was the higher rate of insulin resistance in the MAFLD group (75% vs. 33.8%, *p* < 0.001). After two years of follow-up, NAFLD was found in 46.6% of patients, while 32.6% had MAFLD [68].

An ongoing series of multicentre studies in a Hungarian cohort of patients with CD is currently assessing the body composition and cardiovascular risk-related metabolic parameters (e.g., serum lipids, hemoglobin A1c, homeostatic model assessment index–HOMA-IR, liver enzymes) and metabolic and enteral hormones (leptin, ghrelin, adiponectin) at the diagnosis of CD and following the GFD implementation [70]. The extent of fatty liver disease will be assessed using transabdominal ultrasonography. The results will probably offer supplementary information on the parameters needed to follow up on the signal and prevent the development of SLD and MS [70]. For the moment, we can conclude that patients with CD should undergo regular assessments of metabolic features, including glucose levels, lipid status, serum biomarkers, and liver enzymes. Such evaluations should be repeated over time, based on each patient’s age and risk factors, as well as the current recommendations for the general population [12].

### 3.2. Insulin Resistance

A high intake of saturated and trans fats that are contained in a GFD may lead to increased hepatic lipid accumulation. The stocked triglycerides can cause lipotoxicity, thus leading to hepatocyte injury, inflammation, and activation of stellate cells, which contribute to liver fibrosis. High-glycemic index foods consumed within a GFD can cause rapid increases in plasma glucose levels, triggering hyperinsulinemia, promoting liver lipogenesis, and inhibiting fatty acid oxidation [16,80]. Insulin resistance involves both the peripheral tissue and the liver, which leads to increased hepatic fat accumulation and inflammation, two critical features of MASLD [81].

Insulin resistance is often accompanied by chronic, low-grade inflammation. Inflammatory cytokines, such as Tumor Necrosis Factor (TNF) α and IL-6, disrupt the insulin signaling pathways, exacerbating hepatic insulin resistance and promoting steatosis, liver injury, and fibrosis [81].

### 3.3. Gut Microbiota Alterations

A GFD can lead to changes in the gut microbiota by reducing beneficial bacterial species and increasing harmful ones. The association between CD and SLD might be rooted in the increased permeability of the intestinal mucosa, also known as the “leaky gut,” and the small intestinal bowel overgrowth (SIBO) caused by dysbiosis [82,83]. The gut–liver axis malfunction allows endotoxins and other microbial products to enter the bloodstream more quickly. The influx of endotoxins such as lipopolysaccharides (LPS) reaches the liver, triggering inflammatory responses and promoting liver inflammation, fat accumulation, and fibrosis [84]. As the impaired intestinal barrier and altered microbiota also represent crucial factors for the SLD’s development and progression, a potential pathogenic connection is plausible [85]. Kamal et al. supported this hypothesis by finding that patients with both NAFLD and CD experienced suboptimal and slower histological intestinal improvement after adhering to a GFD compared to patients with CD alone [86].

The gut-liver axis hypothesis postulates that translocation of luminal microbiota or microbial products through a weakened intestinal barrier initiates liver disease by triggering an immune response. This response then causes progressive liver injury and fibrosis by activating the hepatic stellate cells [87,88]. Mechanisms like the decreased production of short-chain fatty acids (SCFA), an altered bile acid pool, and a decreased activation of the farnesoid X receptor (FXR) in the distal small intestine by bile acids seem to be associated with impaired intestinal barrier function [80,89]. In CD, protective bacteria such as *Bifidobacteria, Firmicutes*, *Lactobacilli,* and *Streptococceae* are found in lower amounts compared to healthy controls, while harmful Gram-negative bacteria such as *Bacteroides*, *Bacterioidetes*, *Bacteroides fragilis*, *Prevotella*, *E. coli*, *Proteobacteria*, *Haemophilus*, *Serratia*, and *Klebsiella* are more prevalent [90,91,92]. Notably, an increase in *Bacteroidetes* and other harmful bacteria may also contribute to the development of NASH [93]. Therefore, dysbiosis can disrupt metabolic processes and increase the risk of metabolic diseases, including MASLD [94]. Additional research is needed to ascertain whether this risk is due to the GFD alone or if autonomous, persistent gut-liver axis alterations play a role. Treatments targeting the intestinal barrier dysfunction in chronic liver disease and CD are still lacking.

Recent microbiome studies in CD also emphasize the importance of broader HLA screening, as some bacteria have been associated with cases of CD without classical HLA risk alleles. Incorporating knowledge of the significance of classical HLA associations into risk assessment algorithms could be beneficial, especially when considering a wider range of HLA haplotypes across different populations [95].

Imperatore et al. searched for predictive factors for hepatic steatosis and MS in 301 newly diagnosed CD patients at enrollment and after one year of GFD. After one year of following a GFD, it was found that 112 patients developed hepatic steatosis diagnosed by ultrasonography (25.9% vs. 37.2%; *p* < 0.01), and 72 patients developed MS (4.3% vs. 23.9%; *p* < 0.001). The researchers concluded that almost 33% of the subjects were exposed to proton pump inhibitors (PPI), which was associated with the occurrence of MS (OR 22.9; *p* < 0.001) along with high BMI values at baseline (OR 10.8; *p* < 0.001). PPI exposure (OR 9.2; *p* < 0.001) and HOMA-IR (OR 9.7; *p* < 0.001) also appeared to be associated with hepatic steatosis. Therefore, PPI usage appears to increase the metabolic risk GFD induces in CD and should be limited in these patients [96].

### 3.4. Persistent Inflammation

Chronic high caloric intake results in the expansion of the adipose tissue. Adipose cells, especially those belonging to visceral fat, release pro-inflammatory cytokines such as TNF-α or IL-6 and adipokines such as leptin, which promote insulin resistance and inflammation. These pro-inflammatory mediators then contribute to liver inflammation and fibrosis, exacerbating MASLD [97,98].

A GFD may also alter energy expenditure and metabolic rate, possibly through changes in thyroid function or other hormonal pathways. These alterations can further contribute to weight gain and the metabolic burden on the liver, promoting the development of MASLD. Even while on a GFD, some individuals with CD experience ongoing systemic inflammation due to immune system dysregulation. This can be linked to elevated levels of inflammatory cytokines and immune cell activation [99,100]. Chronic systemic inflammation perpetuates liver inflammation and fibrosis, further contributing to the progression of MASLD [71].

### 3.5. Dietary Missteps

Adherence to the GFD can be compromised by accidental or intentional gluten consumption. Inadvertent intake of gluten products may result from inadequate knowledge or uncontrollable contamination, such as when eating outside the home [101]. In individuals with CD, accidental gluten intake can cause acute inflammatory responses mediated by gliadin peptides that trigger an immune response in the gut and perhaps systemically. Repeated gluten exposure exacerbates systemic and hepatic inflammation, further contributing to the progression of liver disease. Despite best efforts, accidental ingestion of gluten can also occur due to cross-contamination or mislabeling of food products [28]. Non-adherence to the GFD seems to contribute to weight loss, as adherence may play a significant role in weight gain among CD patients. Additionally, the duration of a GFD was identified as a potential variable influencing changes in BMI [60]. A small number of patients can have a celiac crisis or refractory CD with recurrent symptoms (e.g., anemia, diarrhea, weight loss) and persistent duodenal atrophy despite strict adherence to the GFD [102,103].

Since the availability and prices of gluten-free foods have increased, CD and GFD could disproportionately impact poor socioeconomic cohorts, who would resort to buying lower-quality gluten-free foods in more affordable markets, hence the metabolic changes [104]. Without proper dietary guidance, individuals may consume high amounts of processed, nutrient-poor gluten-free foods, leading to an imbalanced diet that is rich in sugars and fats but low in essential nutrients. Such a dietary pattern will promote obesity, insulin resistance, and inflammation, all of which contribute to the pathogenesis of MASLD (Figure 1) [3].

## 4. Treatment Strategies for Preventing MASLD in Celiac Patients

The nutritional adequacy of the GFD has long been controversial and debated. Understanding the beneficial and adverse effects of GFD in celiac patients is primordial for developing targeted health and nutrition strategies to reduce the risk of MASLD in this population. Besides maintaining a safe gluten intake (below 10–50 mg/day), a proper GFD must be nutritionally balanced and meet all energy and nutrient requirements to prevent deficiencies and ensure a healthy lifestyle [105]. Regular monitoring and comprehensive dietary management are crucial to managing these risks effectively. Further research is needed to fully elucidate mechanisms associated with the evolution towards liver steatosis and optimize preventive and therapeutic approaches for MASLD in CD patients.

The European Society for the Study of Coeliac Disease (ESsCD) Guidelines for Coeliac Disease and Other Gluten-Related Disorders have formulated some strong recommendations supported by high or medium evidence for nutritional therapy in CD [4]. Patients with CD must be included in a dedicated disease program managed by a multidisciplinary team in order to benefit from high-quality care that can improve their quality of life [106]. Adherence to a GFD must be monitored regularly. This involves checking clinical status, including anthropometric parameters, conducting dietary assessments using specialized questionnaires, assessing serology parameters, and potentially performing follow-up duodenal biopsies [107]. All celiac-specific antibodies are influenced by gluten exposure, and a decrease in their titer from baseline is expected within weeks to a few months of adhering to a strict GFD. A positive anti-tissue transglutaminase antibody result after starting a GFD suggests ongoing, even minimal, gluten ingestion and is linked to persistent enteropathy. Gluten immunogenic peptides in stool or urine can also indicate ongoing gluten ingestion [108].

As metabolic complications of long-term GFD are possible, strict counseling on diet optimization is needed, given the inadequate nutritional quality of the available gluten-free food products compared to gluten-containing products [66,71]. This can be achieved by increasing physical activity, reducing caloric intake, and minimizing the proportion of highly refined sugars and saturated fats. Such nutritional strategies include the enrichment of unprocessed, naturally gluten-free foods and the inclusion of more fiber, vitamins, and micronutrients [77,109]. Patients should be advised to eat a high-fiber diet that includes whole-grain rice, corn, potatoes, and plenty of vegetables. Oats seem to be well tolerated by most CD patients; the dietary intake should be supervised, and patients should be monitored for potential side effects [4]. Gluten-free wheat flour has proven to be the best solution, showing superior nutritional profiles compared to traditional gluten-free alternatives. These powders usually have a high protein content, good amino acid profiles, and provide enough dietary fiber [110]. Other fake grain flours, such as amaranth, chickpea, chia, and quinoa, have also become popular. These pseudocereals are free of gluten proteins and provide high-quality protein, cholesterol-lowering, and glycemic control effects, as well as enough carbohydrates [110,111]. Testing for micronutrient deficiencies like iron, folic acid, vitamin D, vitamin B6, and B12 in newly diagnosed CD patients and after following a GFD will raise awareness of the possibility that some patients need specific supplementation [4].

Key studies on supplementing a GFD with probiotics such as *Bifidobacterium* and *Lactobacilli* have shown some potential to restore the gut microbiota composition and pre-digest gluten in the intestinal lumen [92,112]. This supplementation may reduce inflammation associated with gluten intake, decrease intestinal permeability, and lower cytokine and antibody production [113]. Since gut microbiota alterations may represent one of the environmental factors involved in CD pathogenesis, probiotics may be useful as a primary prevention method for individuals at high genetic risk for CD [114]. The gut microbiome may even serve as a forthcoming therapeutic target to address the pathogenesis and treatment of concurrent MASLD [115].

Optimal care for the patient with MASLD requires a multidisciplinary approach. Most patients are seen in primary care, where optimized management of medical comorbidities, such as T2DM, hypertension, or obesity, would probably also have beneficial effects on MASLD. Risk stratification should identify at-risk patients among those in this setting. The gastroenterologist’s responsibilities encompass a comprehensive liver risk stratification, the exclusion of other liver diseases, and the focus on liver therapy [14]. Close communication between gastroenterology/hepatology specialists and the primary care medical team would facilitate the multidisciplinary management of metabolic comorbidities and the prioritization of drugs or interventions that can also provide liver benefits. All patients should benefit from a dietary and nutritional assessment and a well-established plan for regular follow-up, independent of the visits to gastroenterology/hepatology specialists. More specialized measures for obesity management, including referral to bariatric surgery, trained psychology specialists, and additional cardiology, lipid, or metabolic support, should be implemented on an individualized basis [14,38].

Patient-specific dietary advice should help CD-MASLD patients maintain an appropriate nutritional intake and reduce the risk of long-term liver-related events. Some of the key recommendations to improve diets include increased consumption of naturally gluten-free foods such as vegetables, fruits, legumes, nuts, seeds, and pseudocereals; choosing the right sources and amounts of good carbohydrates and proteins; and using gluten-free products enriched with micronutrients and vitamins (Figure 2) [110,116]. An ongoing exploration of the correlation between CD, GFD, and MASLD will allow enhanced comprehension and more adequate management of these significant and often associated health issues.

## 5. Conclusions

The pathway leading to the development of MASLD in patients with CD, either before treatment or after GFD is implemented, appears to be multifactorial, including nutritional imbalances, increased fat and sugar intake, weight gain, insulin resistance, altered gut microbiota, chronic inflammation, and poor dietary intake. Addressing these factors through regular screening, comprehensive nutritional management, and sustained lifestyle changes may help reduce the risk of MASLD in this population. At present, the effect of GFDs on MASLD development in CD patients is not yet fully understood. Further research is needed to explore this relationship, as a GFD may theoretically increase the risk of metabolic syndrome features due to its specific ingredient components. More research is needed to better understand these relationships and to identify effective prevention and treatment strategies.

## Figures and Tables

**Figure 1 nutrients-16-02008-f001:**
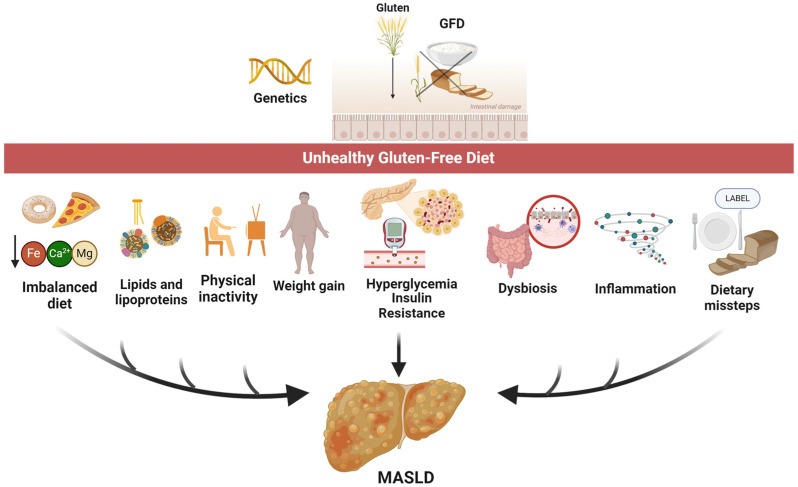
Potential pathways leading to MASLD in patients with CD following an unhealthy GFD.

**Figure 2 nutrients-16-02008-f002:**
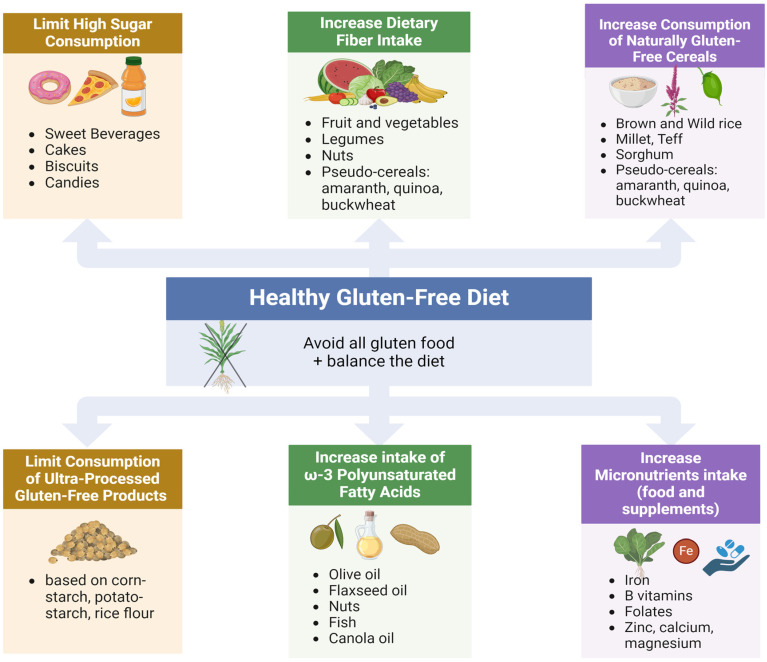
A healthy GFD aimed to prevent MASLD in CD patients.

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
