# Peer review of "Celiac Disease, Gluten-Free Diet and Metabolic Dysfunction-Associated Steatotic Liver Disease"

_nutrients, 2024, doi:10.3390/nu16132008_

Round 1

Reviewer 1 Report

Comments and Suggestions for Authors

The work is very well written, the correct construction, I have only minor comments:

Row 38 – there are not 75% of genetically predisposed people - I don't fully understand. Genetic predisposition is about 30-40%  of them – explain what does 75% mean, what does it refer to? Because I don't think it's for inheritance either

Row 40 – people with CD are not always intolerant to oats

Line 72 – no explanation of the abbreviation MASLD – is in the abstract, but it should also be in the text, Line 125 – introduces only the abbreviation – Metabolic dysfunction-associated Steatotic Liver Disease (MASLD).

Wiersz 93-96 The diagnosis of CD requires a positive serology for IgA anti-transglutaminase 2 93 and anti-endomysial antibodies, along with evidence of villous modifications on a small 94 intestine biopsy, such as varying degrees of villus atrophy, mucosal inflammation, and 95 crypt hyperplasia – in this sentence you are misleading – because the basis of diagnosis is ttg IgA and Total IgA. In the diagnostic step form, biopsy – always in adults, and in children with tTG above 10xcut off, we can think about resigning from the biopsy and only then endomysium

The sentence in lines 43 and 96 – is repeated

Line 180 NHANES abbreviation no expansion

Reviewer 2 Report

Comments and Suggestions for Authors

This work is a review and therefore has nothing original, but it appears useful for a complete understanding of the relationship between celiac disease, gluten-free diet and liver damage.

The only observation I feel like making is that it seems too long to me. Surely all the concepts expressed can be told with fewer words. This would make reading easier.

Reviewer 3 Report

Comments and Suggestions for Authors

The study explores the relationship between celiac disease (CD), adherence to a gluten-free diet (GFD), and the development of metabolic dysfunction-associated steatotic liver disease (MASLD). It highlights the dual role of GFD in managing CD symptoms while potentially increasing the risk of hepatic steatosis and other cardiometabolic risk factors. The review delves into the causative factors linking CD and hepatic injury, presenting both as extraintestinal manifestations of CD and as side effects of GFD, and discusses potential therapeutic strategies for these patients.

Congratulations to the authors on a thorough and well-written review. The paper provides a comprehensive overview of the complex interplay between celiac disease, gluten-free diet, and metabolic dysfunction-associated steatotic liver disease. The detailed analysis and synthesis of current evidence make a significant contribution to understanding the multifaceted impact of CD and GFD on liver health.

To further enhance the discussion, I suggest including references to the rapid weight loss and malnutrition associated with celiac crisis - https://doi.org/10.1053/j.gastro.2020.07.054 - or refractory celiac disease type 2 - https://doi.org/10.1590/1806-9282.67.02.20200618 - as causes for steatosis. This additional context could provide a more nuanced understanding of the pathways leading to liver steatosis in patients with CD, thereby enriching the existing narrative

Comments on the Quality of English Language

English language is fine.
